# Choice of Piezoelectric Element over Accelerometer for an Energy-Autonomous Shoe-Based System

**DOI:** 10.3390/s24082549

**Published:** 2024-04-16

**Authors:** Niharika Gogoi, Yuanjia Zhu, Jens Kirchner, Georg Fischer

**Affiliations:** 1Department of Computer Science, Durham University, Upper Mountjoy Campus, Stockton Road, Durham DH13LE, UK; niharika.gogoi@durham.ac.uk; 2Institute of Technical Electronics, Friedrich-Alexander-Universität Erlangen-Nürnberg, 91058 Erlangen, Germany; yuanjia.zhu@fau.de (Y.Z.); jens.kirchner@fau.de (J.K.); 3Faculty of Information Technology, University of Applied Sciences and Arts, 44227 Dortmund, Germany

**Keywords:** accelerometer, piezoelectric, gait, step-count, activity recognition

## Abstract

Shoe-based wearable sensor systems are a growing research area in health monitoring, disease diagnosis, rehabilitation, and sports training. These systems—equipped with one or more sensors, either of the same or different types—capture information related to foot movement or pressure maps beneath the foot. This captured information offers an overview of the subject’s overall movement, known as the human gait. Beyond sensing, these systems also provide a platform for hosting ambient energy harvesters. They hold the potential to harvest energy from foot movements and operate related low-power devices sustainably. This article proposes two types of strategies (Strategy 1 and Strategy 2) for an energy-autonomous shoe-based system. Strategy 1 uses an accelerometer as a sensor for gait acquisition, which reflects the classical choice. Strategy 2 uses a piezoelectric element for the same, which opens up a new perspective in its implementation. In both strategies, the piezoelectric elements are used to harvest energy from foot activities and operate the system. The article presents a fair comparison between both strategies in terms of power consumption, accuracy, and the extent to which piezoelectric energy harvesters can contribute to overall power management. Moreover, Strategy 2, which uses piezoelectric elements for simultaneous sensing and energy harvesting, is a power-optimized method for an energy-autonomous shoe system.

## 1. Introduction

The study of wearable devices is a trending research topic, owing to their non-invasive approach and ability to deliver continuous, real-time monitoring of physical activities. Wearable devices have existed for several decades—the first being the Holter monitor (1962), which records the heart’s rhythm [1,2]. Over the years, the discovery of new sensor materials, cutting-edge fabrication techniques, robust models for sensor data analysis, faster computation power, and innovative power management solutions have evolved the capabilities of wearable devices, enhancing their reliability and performance [3,4].

A wearable device integrates several components, such as sensors, microcontroller units, transceivers, and power supply. The sensors are used to monitor human physiological parameters, such as heart rate and blood pressure, as well as biomarkers, such as glucose, sodium, and other minerals [5,6,7,8]. The microcontroller unit collects, processes, and interprets the sensor data. The transceivers transmit useful health information obtained from sensor data to a physician, caretaker, sports trainer, rehabilitation therapist, or any concerned individual, depending on the area of application. The power supply is another key element needed to operate the entire system. All these components determine the efficient working of a wearable device. Recently, research on wearable devices has advanced into a new era of the Internet of Health Things (IoHT), a cloud platform that integrates various sensor systems with information technology, offering data analysis, storage, and immediate action commands for critical situations [9,10,11,12,13,14]. To name a few, the IoHT has been implemented to enhance the quality of life for elderly patients, provide local communities with access to emergency medical assistance, and enable remote patient monitoring using smartphones [15,16].

### 1.1. Shoe-Based Human Gait Acquisition

The mobility or gait of an individual is affected by neurological, orthopedic, and musculoskeletal disorders, such as those occurring due to aging, stroke, diabetes, and injuries. Based on the position of sensors and improved machine learning algorithms for signal processing, gait disorder is used as an indicator of the advancement of illness, the impact of therapy or sports training, and fitness tracking [17,18,19,20,21]. The gait disorder outcome is observed in terms of impaired balance, limping, and frequent falls. Technically, gait disorders result in changing stride length, plantar pressure, and heel and toe strike instants, measured from sensors placed in socks, shoes, or soles [22,23,24]. The most common sensors used in such wearable systems are an accelerometer, gyroscope, magnetometer, and pressure sensor array [25,26,27,28]. Inertial measurement units (IMUs) are quite popular for gait analysis. They can provide several pieces of information, such as specific force, angular rate, and orientation of the subject. To summarize, it involves a combination of an accelerometer, gyroscope, and magnetometer. An IMU or separate accelerometer, gyroscope, or magnetometer is usually placed on the thigh, waist, ankle, or knee, attached in a band [29,30]. In the work of [31], the optimal location and orientation of an IMU sensor on a barefoot were investigated, while [32], explored the placement of a 3-axis accelerometer and gyroscope on top of the shoe to assess patients with Parkinson’s disease. For shoe-based sensor systems, e-textiles, such as smart socks and insoles, show promise for implementing remote gait monitoring [33,34].

The focus of this article is a shoe-based wearable sensor system, with its application in human gait analysis. A shoe is an ideal choice as it can be worn by individuals of all age groups and health conditions. There are several scientific investigations on such systems. The authors in [35] examined gait with a combination of two foot sensors and a mobile app (SmartMOVE) to detect the falls of patients suffering from Parkinson’s disease. One of the earliest forms of research in this area involved a system used for measuring pressure distribution under the foot. It was conducted with 7 force-sensitive resistors (FSRs) to identify between shuffling and walking movements [36,37,38]. Another group developed a system capable of detecting temporal gait parameters with 2 FSRs positioned under the foot to detect falls from elderly people and to find fluctuations in gait patterns [39,40,41]. Multimodal smart soles were developed to investigate gait patterns from different types of sensors for different applications. Lechal used inertial sensors to alert visually impaired individuals while Sensoria used textile pressure sensors to avoid injuries for runners [42,43]. A multimodal electronic textile-based pressure sensor and a low-cost inertial measurement unit, equipped with a three-axis accelerometer, magnetometer, and gyroscope, were integrated into the insole for activity recognition. This setup facilitated long-duration data collection from subjects [44]. Conductive soft pressure sensor arrays molded inside silicon rubber structures were developed with a focus on the comfort of the subject. This design was implemented to sense plantar pressure parameters in both static and dynamic situations [45]. Similar kinds of customizable flexible sensor arrays were proposed in [46,47], where piezo-resistive and capacitive sensors were implemented to acquire plantar pressure distribution. In reference [48], a smart shoe-based platform implemented with a GPS tracker was used to navigate a blind person. Using voice messages, it guided the individual around physical obstacles, fire, water, and potholes. A similar kind of idea has also been implemented in boots for real-time monitoring of autistic people in the absence of a caretaker [49]. The authors in [50] proposed a power-optimized solution for gait monitoring from two accelerometers with an improved algorithm.

### 1.2. Power Management of Smart Shoe Insole

With the growth of wearable sensor-based devices, there is growing concern about battery management. Most conventional wearable devices are operated by batteries, which have a limited lifespan and need replacement. The replaced batteries, which contain toxic substances, are disposed of in the environment [51]. This is alarming and contributes to land pollution. Ensuring a reliable and constant power supply in a sustainable approach is a prerequisite of modern wearable devices. Therefore, researchers and scientists are shifting their focus to alternative power supplies. Ambient energy harvesting options, such as piezoelectric, thermoelectric, triboelectric, and electromagnetic methods, are possible ways to increase the battery’s lifetime and lower the rate of its disposal [52,53,54,55,56]. The human body has the potential to contribute toward renewable energy, making wearable devices like smart shoes self-powered. Theoretical calculations estimate that body heat, breathing, and arm movements can generate 2.8–4.8 W, 0.83 W, and 60 W [57], while ambulatory footfall generates 20 W [55,58]. Only a percentage of the estimated power can be converted through thermoelectric, piezoelectric, or inertial energy harvesters to operate wearable devices. For example, measurements showed that a piezoelectric stack actuated by hydraulic amplifiers could generate 150–700 mW while walking and 600–2500 mW while jogging [59]. Depending on the harvester design and position, piezoelectric elements can harvest from a few μW to mW [60,61,62,63]. The temperature gradient and ionic imbalance in sweat also represent promising options for batteryless applications [64,65,66,67]. The scope of this article is piezoelectric energy harvesting, taking advantage of vibration induced by human motion.

The mechanical vibrations generated while walking, breathing, and muscle movement are promising sources of energy. Human walking and vehicle movement have the potential to harvest energy via piezoelectric elements embedded in roads and shoes [68]. Out of the various options, this work explores piezoelectric elements for energy harvesting. A group of researchers used a lead zirconate titanate (PZT) layer mounted over a stainless steel structure with a spring to generate the effects through compression and relaxation [69,70]. In one of their articles, they used this structure to drive a circuit to switch on LED in the shoes of workers. They also proposed a self-powered system without the use of any battery, such that it generated sufficient power to drive the transmitter after 8 steps. In another article [71], the authors investigated mechanical and electrical measures generated at different positions of various types of shoes. The wedge-heel type of shoe generated a high open-circuit voltage and a high short-circuit current at the toe strike, while the block-heel type of shoe showed the same at the heel strike. It showed that larger-sized piezoelectric elements generated more energy, and increasing the speed of walking increased the force exerted by the foot. Another energy harvester with 2 piezoelectric cantilever beams—each with 2 ratchets, a gear, an arc rack, a spring, 2 axles, 4 bearings, and 2 one-way bearings—was proposed in [72]. The main objective of this arrangement was to increase high power peaks in each gait cycle. The authors in [73] proposed a hexagon structure with polylactide thermoplastic using parallel links of piezoelectric harvesters. This design optimization increased the output to 1.29 mW of power. The sole optimization, e.g., stainless steel in [69] and parallel links in [73], improved the harvested output but made the shoes uncomfortable to wear and move around. In another work, a maximum power point tracking (MPPT) circuit interface was implemented to improve the harvested power from daily walking for gait monitoring insole [74].

### 1.3. Proposed Work

In most articles, sensing and energy harvesting are separately investigated, whereas, in our work, we investigate both sensing and energy harvesting embedded together in a shoe sole. In conventional human gait acquisition applications, an accelerometer is a common choice for collecting gait information while an array of piezoelectric elements is used to generate a plantar pressure map. The proposed work is different from the conventional approach. It explores the potential and limits of a piezoelectric element for gait acquisition and activity recognition. Two strategies, namely Strategy 1 and Strategy 2, are presented in this article based on the sensor choice. The accelerometer is chosen in Strategy 1 and the piezoelectric element is chosen in Strategy 2 to acquire sensing data. The same low-cost piezoelectric elements are used as ambient energy harvesters embedded in shoes to generate energy while walking. This research investigates the extent to which such harvesters can support the operation of smart shoe applications and reduce dependency on batteries for a specific set of sensor functionalities.

For any system-based application—an energy-autonomous shoe, in our case—several elements coordinate to achieve the objective. The highlight of this work is to achieve simultaneous sensing and energy harvesting with a comprehensive approach. Low-power electronic components are chosen and the sole design is improved for better electromechanical conversion. The comfort of a shoe sole and the affordable cost of the components are also taken into consideration for experimental demonstration. This article compares both strategies, in terms of their power consumption and sensing accuracy for a smart shoe application. The stage-wise power consumption study of each component is important for an energy-autonomous operation. For both strategies, this article provides an overview of a detailed understanding of the power consumption in sensor data acquisition, processing, and transmission. The novelty is that the piezoelectric elements in Strategy 2 offer a power-optimized method and an alternative accelerometer choice for lifestyle wearables. This approach is particularly appealing given the growing interest in utilizing ambient energy harvesters to extend battery life and slow down the rate of battery disposal.

## 2. System Design

Figure 1 shows the block diagram of the two strategies. It constitutes (1) a sensor (accelerometer in Strategy 1 and piezoelectric element in Strategy 2), (2) a microcontroller unit, (3) a data transmission module, (4) an array of five piezoelectric energy harvesters, each with a rectifier, (5) a voltage regulator, and (6) a switch. The rectifier, voltage regulator, and switch together comprised the interface circuit. A voltage divider is added in Strategy 2 (check Figure 1b) as protection against voltage spikes. A battery is also added as a backup in both strategies. The impact of the battery in both strategies will be discussed at the end of the article. Low-power components are chosen for the experiment to ensure low current consumption and extend battery life.

### 2.1. Sensor

A 3-axis MEMS accelerometer ADXL362 (denoted as ACC) [75] is placed on top of a shoe on a circuit board, such that its Y-axis is aligned with the forward walking direction of the subject. The measurement range of ACC is ±2 g, ±4 g, and ±8 g. The sampling frequency in this design is set to the lowest value of 12.5 Hz. The upper limit of the FIFO buffer is set to 300 samples, which implies that the ACC uploads data every 8 s for each axis. It is switched between two operating modes—measurement and wake-up. The wake-up mode reduces power consumption until an activity is detected based on a defined threshold. If an activity is detected, the ACC sends an ‘interrupt’ to the microcontroller to record the measurement via the SPI interface.

Figure 2 shows the piezoelectric element implemented in this study. It was purchased from Murata Electronics [76]. It is a brass disc with a PZT layer on top of it and coated with a silver electrode. The same element was used for sensing and energy harvesting. It was placed under the foot to experience the pressure while walking. In further discussion, it will be denoted as PZT.

### 2.2. Microcontroller Unit (MCU)

The MSP430FR5947, manufactured by Texas Instruments, is used as a microcontroller unit due to its ultra-low power system architecture [77]. It operates at 1.8 to 3.6 V and consumes a maximum of 1.3 μA under normal operation in a low-power mode. The power consumption is minimized by disabling registers on the MCU, supporting several low-power modes, according to the needs of different applications. It will further be denoted as MSP430.

### 2.3. Data Transmission Module

The data obtained from the sensors were transmitted to the MSP430 by Bluetooth DA14531 [78]. It is the world’s smallest and lowest-power Bluetooth 5.1 system-on-chip (SoC). It operates at a battery voltage range from 1.1 to 3.3 V, with a receiver sensitivity of −94 dBm. Its programmable transmit output power ranges from −20 dBm to +2.5 dBm. The current consumption of the transmitter is 3.5 mA at 0 dBm and that of the receiver is 2.2 mA at an input voltage of 3 V. It will be denoted as ‘BLE Broadcaster’ in further discussions.

The data transmission starts as soon as the Bluetooth interrupt is triggered and the timer is set. It starts to broadcast the data received from the MSP430 through the UART bus at a broadcast interval of 1.5 s. When the timer overflows, the Bluetooth chip turns off all the peripherals, except the wake-up ‘interrupt’, and enters sleep mode to ensure minimum power consumption.

### 2.4. Piezoelectric Energy Harvesters

The same element in Figure 2 is used as the energy harvester. A silicone rubber structure in the shape of a sole with dimensions of 95 mm × 240 mm is designed to host the harvesters. The piezoelectric elements are placed on the layout as shown in Figure 3. The numeric denotations 1, 2, 3, 4, and 5 are the positions of the energy harvesters while S is the position of the PZT used for sensing purposes. The position of each element is chosen in such a way that, as the foot strikes the ground, all of them also strike the ground. Detailed information about the sole design can be found in Section 5.2.2 of [79]. The insole design is flexible and comfortable but unbreakable while walking. It also ensured the protection of the solder joints of PZT elements and improved their durability.

### 2.5. Interface Circuit

The interface circuit is composed of three parts: rectifier, voltage regulator, and switch.

#### 2.5.1. Rectifier

A full bridge standard energy harvesting circuit (SEH) is used to rectify the piezoelectric voltage, $V_{p}$, generated while walking. It consists of 4 diodes arranged as a wheat-stone bridge, a filter capacitor, $C_{L}$, and an equivalent resistance of the terminal load, $R_{L}$, as shown in Figure 4. In our published article [80], a PZT element was connected to the SEH circuit, and 5 of such SEH circuits when connected in parallel gave more output power than other nonlinear rectifier circuits. Therefore, we implemented the full bridge rectifier circuit in this work.

#### 2.5.2. Voltage Regulator

The nano buck–boost converter LTC3331 is used to regulate the harvested output at a desired voltage [81]. It also acts as a battery charger and has a low battery disconnect function that protects the battery from deep discharge. The harvested energy is used to power the MSP430 and charge the battery. When harvested energy is not available, the LTC3331 uses the battery as the source to drive the load.

#### 2.5.3. Switch

Figure 5 shows a switch circuit, which consists of an R-C network and an N-channel MOSFET. It is implemented to let the LTC3331 attain its rated output voltage first and then drive the load. The power good ($PG_{V_{OUT}}$) pin of the LTC3331 decides when to turn on the MOSFET. Until the output $V_{OUT}$ of LTC3331 is below its rated voltage, $PG_{V_{OUT}}$ is low. $PG_{V_{OUT}}$ is high when the $V_{OUT}$ of the LTC3331 reaches its rated voltage. This switch ensures that the system is cold-started on its own without any manual contribution.

### 2.6. Final System Design

The PCB schematics and final boards are shown in Figure 6. The seven parts of the board are highlighted in both the sub-figures. The proposed sole layout placed inside the shoe is shown in Figure 7.

## 3. Sensor Data Acquisition

This section is divided into 2 subsections: Strategy 1 and Strategy 2.

### 3.1. Strategy 1: Accelerometer plus BLE Broadcaster

The communication in Strategy 1 is shown as block diagrams in Figure 8. It only has 2 interrupt signals: overflow and BLE. The overflow interrupt of ACC implies that its FIFO sample register has collected 100 sets of samples for each of the three axes. At that instant, the data are extracted and processed by MSP430, and after processing, they return to the low-power mode. Then, the MSP430 calls the BLE ‘interrupt’ to send advertising packets, facilitating the transmission of the processed data to the subject’s mobile phone. The MSP430 sends the processed data to the Bluetooth chip through the UART bus.

Before the algorithm of ACC signal processing is defined, different types of activities are captured and graphically presented. The reference axis of acceleration data for activity recognition is presented in the bottom right. In Figure 9, an activity is identified by the peak and time intervals between any two consecutive peaks. Two activities are mainly detected: walking and running. The up climb and down climb are considered as walking, i.e., we assume Figure 9a,c,d as walking. However, the pattern of the Y-axis acceleration changes for the up climb, which we will discuss later. The walk, run, up climb, and down climb actions are based on Y-axis acceleration (as depicted by green graphs). The number of peaks implies the activities. The acceleration values of the run are higher than those of the walk. Therefore, a different threshold is set for Y-acceleration to distinguish between walking and running. It is critical that for every step taken to walk uphill, two consecutive peaks occur as shown in Figure 9d. The peak detection of such consecutive peaks would double the number of steps. As a result, the up climb is detected by 2 conditions—the Y-axis acceleration threshold and the time interval between consecutive peaks. Due to physical movement, additional noise is added to the acceleration results, which affects the accuracy of activity recognition. A threshold value is also set to eliminate noise, such that acceleration peaks lower than this threshold are ignored.

### 3.2. Strategy 2: Piezoelectric Sensor + BLE Broadcaster

#### Gait Signal

A typical gait signal is shown in Figure 10. It is recorded from a piezoelectric element placed on the heel of a shoe. The power graph is calculated based on its intrinsic resistance. There is no other electrical load connected to it. The above measurement is recorded before the elements are embedded in the silicone layout mentioned above. As shown, it has an asymmetric positive spike with a peak power of about 1.2 mW. On the one hand, the gait signal needs to be regulated for a stable operation, which is conducted by the LTC3331. On the other hand, the spikes denote the number of steps taken by an individual. Thus, the same signal provides sensor information and is implemented for energy harvesting.

The communication in Strategy 2 is shown as block diagrams in Figure 11. It discusses the implementation of the piezoelectric sensor in the proposed work. The GPIO port of the MSP430 detects the rising edge and the falling edge. As a piezoelectric element is pressed while walking, the interrupt function detects such an edge of the voltage across the PZT. This triggers an interrupt, which turns on the timer. The timer records the value of the instant when a peak is generated and the time interval between two interrupts is stored. Based on the time interval, only run or walk activities are recognized. Unlike the ACC, PZT provides only uniaxial information that is sufficient for the detection of walk-and-run activities. To recognize the jump activity, the force applied on PZT has to be determined. The jump activity recognition from the piezoelectric element depends on the force exerted while walking. The force generated by each activity corresponds to the voltage. The accurate PZT voltage can be measured by the ADC module of MSP430. However, the use of the ADC module is restricted to lower computational power consumption.

### 3.3. Power Calculation

The power measurement is based on the current measured by the Nordic Power Profiler Kit [82] and the operating voltage of the system. The equation followed for the power calculation is as follows:(1)P=V×I,
where the current is obtained from the kit and the voltage is 3.3 V.

### 3.4. Accuracy Calculation

The accuracy of sensor data acquisition is calculated by the ratio of recorded to actual activities, i.e.,
(2)$$Accuracy\left(\%\right)=\left(1-\frac{\frac{|Error_{run}|}{Actual_{run}}+\frac{|Error_{walk}|}{Actual_{walk}}}{2}\right)\times 100,$$

The Error is the difference between the actual and recorded activity. It is divided by 2 because two activities are considered. Actual is the activity manually observed and is considered ground truth for reference.

## 4. Results and Discussion

### 4.1. Power Consumption during Data Acquisition, Processing and Transmission

The power consumption of the proposed system has two phases, namely, (a) Phase 1: data acquisition and processing by MSP430, and (b) Phase 2: data transmission by the BLE broadcaster.

Figure 12 shows the power consumption of the two strategies for the first phase. In Strategy 1, when the ACC collects sufficient samples, the MSP430 extracts data through serial communication and processes it. At this moment, the power consumption reaches a peak at 4.5 mW for a short instant of 0.38 s and then drops (see Figure 12a). The average power consumption of the system for every gait cycle is about 0.56 mW. In Strategy 2, the peak power consumption is about 1.3 mW, and the average power for the entire 50 s period is 0.6 mW (Figure 12b). The distributed pressure under the foot generates noise. Each noise wakes up the MSP430 from the low-power mode, which results in a higher average power of the PZT than the ACC. However, the peak power consumption in Strategy 2 is less than one-third of that of Strategy 1.

Figure 13a shows the power consumption of the data transmission by the BLE broadcaster. As the same data transmission module is implemented, the power consumption for this function remains the same for both strategies. Its average power consumption is 0.66 mW for the advertising interval of 1.5 s and the peak power consumption is about 7 mW for 0.02 s. The critical situation occurs during the peak instant. If not advertising, the broadcaster is in a sleep state, ensuring low power consumption. Figure 13b shows the power consumption at the advertising moment. The first peak represents the broadcaster that is woken up from the sleep state. The next three peaks occur when it broadcasts data to channels 37, 38, and 39, respectively, after which, it returns to sleep mode. To further lower the power consumption in data transmission, the BLE broadcaster can be dropped and sensor data can be obtained from UART mode.

There is additional power required to operate the ACC, which leads to an increase in the overall power consumption in Strategy 1. The PZT needs no such external power to operate. Its power consumption in Strategy 2 could be further minimized by avoiding the timer but its removal would fail to achieve the activity recognition and perform only step counting like a pedometer.

### 4.2. Accuracy Comparison of Both Strategies

Table 1 and Table 2 show the accuracies of both strategies based on the two types of sensors used. The ACC and PZT identify two different types of activity recognition. Strategy 2 is proposed in a way that the energy harvested while walking can deliver enough output to operate its mentioned functionalities. The accuracy of PZT is about 10% less than that of ACC. Due to dispersive plantar pressure distribution, the algorithm misinterprets noise and detects it as the peak.

### 4.3. Output of Interface Circuit

Figure 14 shows that the LTC3331 output achieves the desired voltage around the same instant in both strategies. Taking advantage of a Power Good signal in the switch circuit, the MOSFET ensures that the LTC3331 achieves the desired voltage before driving the load.

Figure 15 shows the LTC3331 output without an additional battery. Figure 15a shows that in Strategy 1, after the 8th step, the LTC3331 output reaches an unstable voltage of peak 2.8 V and then drops to 0.9 V. As it fails to achieve the desired 3.3 V and lower operating limit of the MSP430, i.e., 1.8 V, it is unable to initialize itself and the ACC peripherals. However, the collection of gait data is possible in Strategy 2. Figure 15b shows that the LTC3331 output achieves the desired value of 3.3 V and drops to 1.8 V. As 1.8 V is the lowest operating voltage of the MSP430, Strategy 2 can work with the fluctuating voltage. Therefore, Strategy 2 can perform two types of activity recognition without requiring an additional battery. It is clear from Figure 14 and Figure 15 that a battery acts as a backup storage capacitor. It provides a stable voltage supply to the system and possibly improves the accuracy of sensor data acquisition.

Figure 10 shows that one element produces about 1.2 mW of peak power and 0.7 mW of average power. Therefore, with 5 of such elements, about 5–6 mW of peak and 3.5 mW of average power can be generated. This is sufficient for the peak power consumption of Phase 1, i.e., data acquisition and processing by MSP430, which is only 4.5 mW in Strategy 1 and 1.3 mW in Strategy 2 (ref Figure 12). In the data transmission phase, a peak power demand of 7.3 mW is required for 0.02 s (ref Figure 13). It is during this time that the battery is needed for a short period. The interface circuit is designed to utilize the extra harvested energy to charge the battery. Consequently, the battery does not require an external power supply for charging, making the system energy-autonomous. To make it batteryless and energy-autonomous, the data transmission module can be replaced with the UART mode.

## 5. Conclusions

In this work, an energy-autonomous smart shoe system with energy harvesting, activity recognition, and wireless connectivity was successfully designed, built, and experimented with. It presents a comprehensive study of stage-wise power consumption, constituting components, the accuracy of sensor data, and the impact of energy harvested from walking. Two strategies were discussed based on the choice of the sensor. Strategy 1 uses ACC as a sensor, which is the most common choice for the gait acquisition system, and Strategy 2 uses PZT for the same. The choice of the sensor determines the power consumption in the data acquisition and processing, as well as the overall accuracy of the obtained data. The power consumption in the wireless data transmission phase is the same for both strategies. It is the most power-hungry part of the proposed design, with a peak power of around 7 mW, making it critical to operate as a batteryless application. To reduce the dependency on batteries or external power supply, low-cost PZT elements are implemented to harvest energy and charge the battery in both strategies.

The comparison between the two strategies is presented in Table 3. It shows the trade-off between accuracy and energy consumption. Strategy 2 opens up an ideal technique to use PZT for simultaneous sensing and energy harvesting. It is apparent that Strategy 1 has a higher average accuracy than that of Strategy 2. However, with the growing demand for wearable and battery disposal concerns, power optimization is a basic necessity of any electronic wearable system design. One attractive feature of Strategy 2 is its low power consumption in data acquisition and processing, which is about one-third of that of Strategy 1. Unlike an ACC, a PZT does not require any external power supply to operate. Another attractive feature of Strategy 2 is the high peak-charging current per step, i.e., 75 μA/step, compared to Strategy 1. This current is used to charge the battery. The power optimization in Strategy 2, by a few μW–mW, contributes to a significant value when billions of such systems are commercialized at a large scale. The use of PZT elements as energy harvesters minimizes the dependency on the external power supply and increases the lifetime of the battery. Nevertheless, it ultimately depends on the preferences of users on whether to opt for accuracy or low power consumption. For trending lifestyle applications, where accuracy is not a significant concern, PZT as a sensor and energy harvester is a promising choice.

The output of piezoelectric elements in the proposed application depends on several factors. One important factor that improves energy harvesting is its electromechanical conversion efficiency, which is determined by the type of material and sole layout, where the elements are embedded. As material synthesis is beyond the expertise of the authors, low-cost, lightweight commercial elements are considered. Within the scope of the authors, a silicon rubber sole layout is designed to ensure the comfort of the subject, and low-power electronic components are chosen. The microcontroller unit (MSP430FR5947) and voltage regulator (LTC3331) implemented in the study have low power consumption. With all these considerations, a power-optimized approach for an energy-autonomous shoe is proposed.

## Figures and Tables

**Figure 1 sensors-24-02549-f001:**
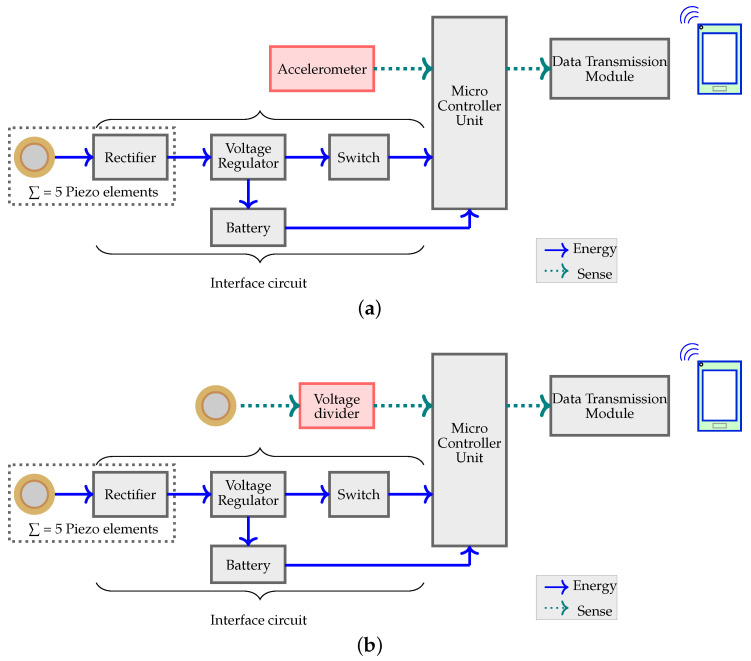
System design: (**a**) Strategy 1 and (**b**) Strategy 2.

**Figure 2 sensors-24-02549-f002:**
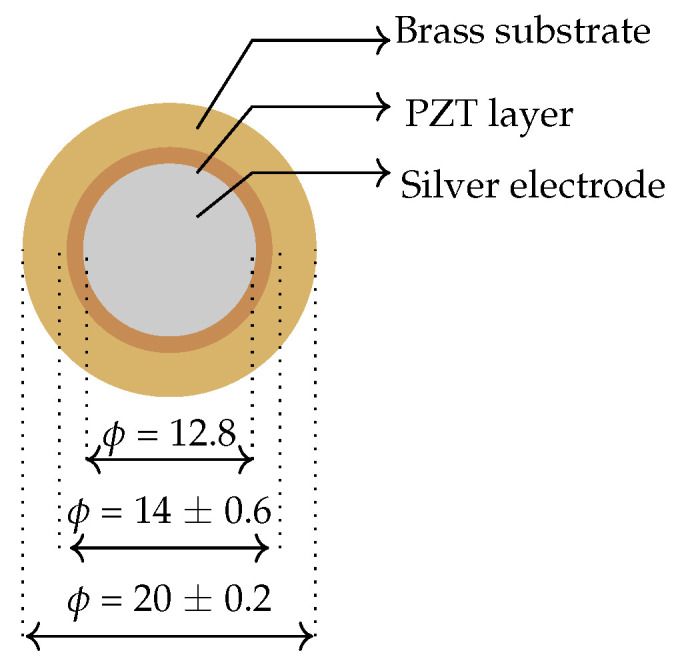
Piezoelectric sensor (all dimensions are in mm).

**Figure 3 sensors-24-02549-f003:**
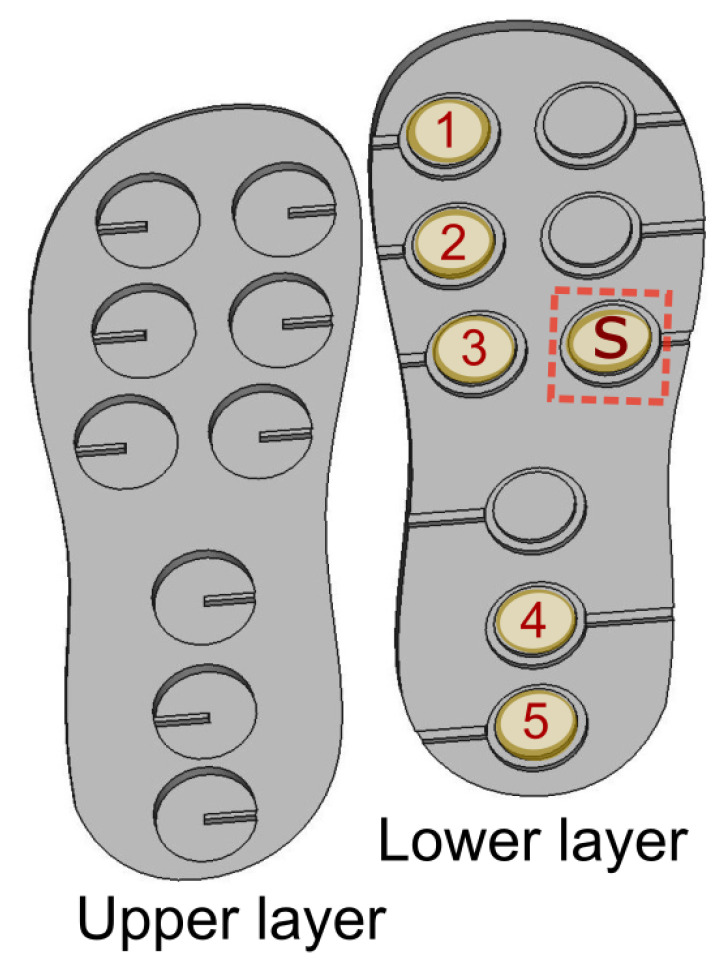
Positions of 6 piezoelectric elements on the sole layout.

**Figure 4 sensors-24-02549-f004:**
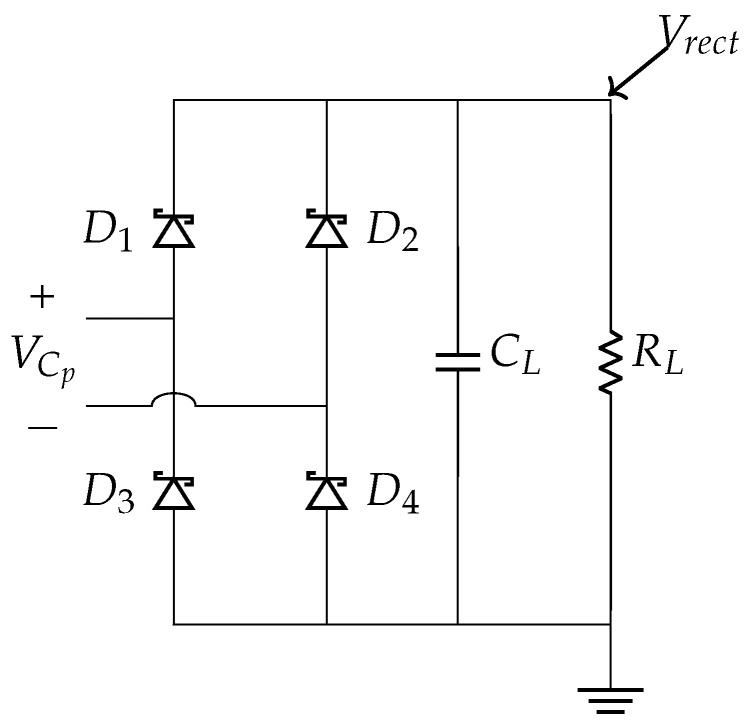
Rectifier circuit.

**Figure 5 sensors-24-02549-f005:**
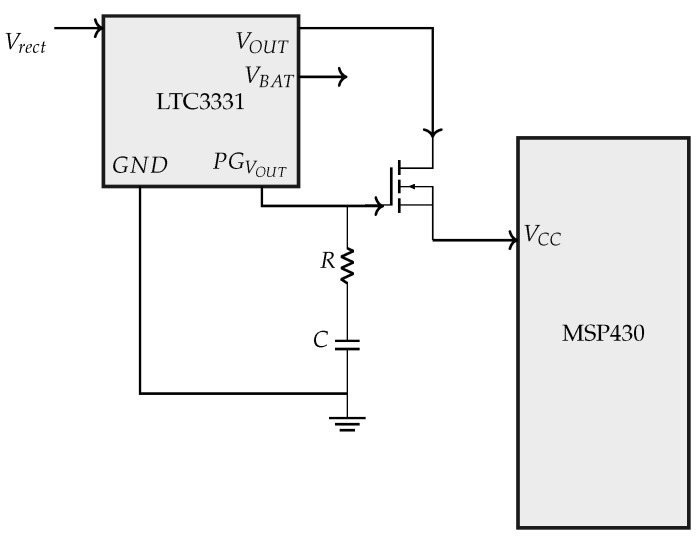
Switch circuit.

**Figure 6 sensors-24-02549-f006:**
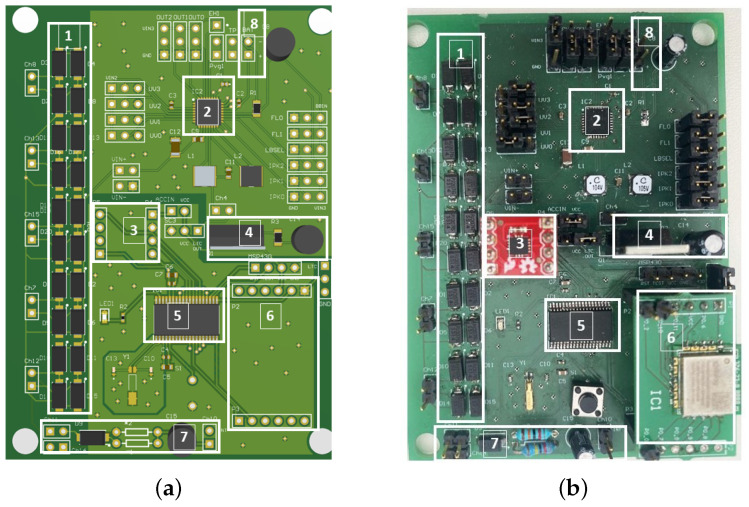
(**a**) PCB schematic and (**b**) final board. 1: Rectifier circuit for 5 PZT elements; 2: voltage regulator; 3: accelerometer; 4: switch circuit; 5: microcontroller unit; 6: data transmission module; 7: voltage divider; 8: batter pin.

**Figure 7 sensors-24-02549-f007:**
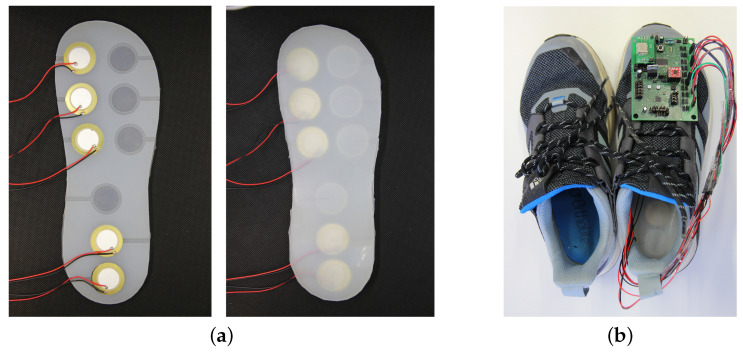
(**a**) Final sole layout and (**b**) Real prototype.

**Figure 8 sensors-24-02549-f008:**
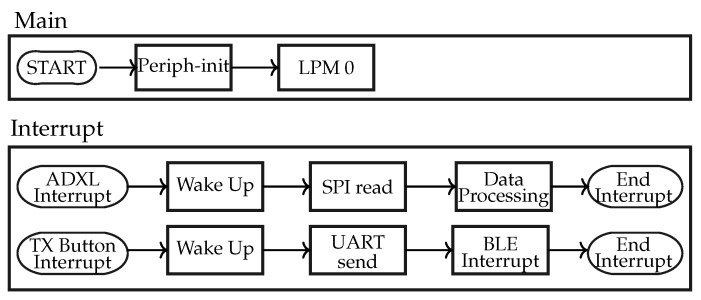
Block diagram: Strategy 1.

**Figure 9 sensors-24-02549-f009:**
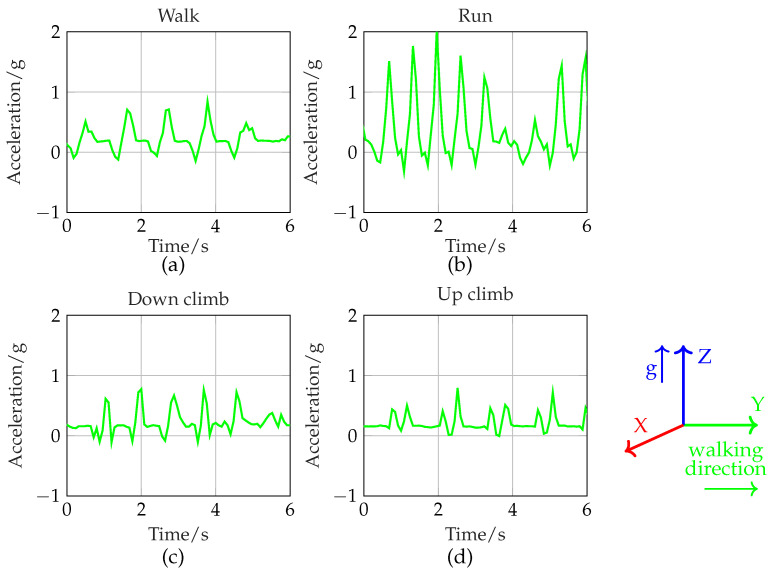
Y-axis acceleration for (**a**) walk, (**b**) run, (**c**) down climb, and (**d**) up climb.

**Figure 10 sensors-24-02549-f010:**
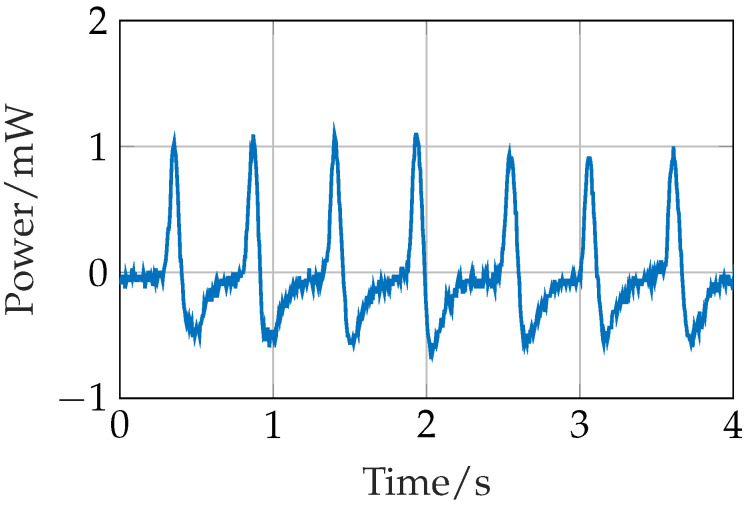
A typical gait signal from one piezoelectric element.

**Figure 11 sensors-24-02549-f011:**
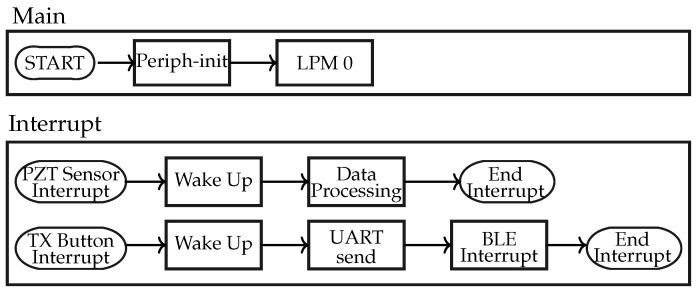
Block diagram: Strategy 2.

**Figure 12 sensors-24-02549-f012:**
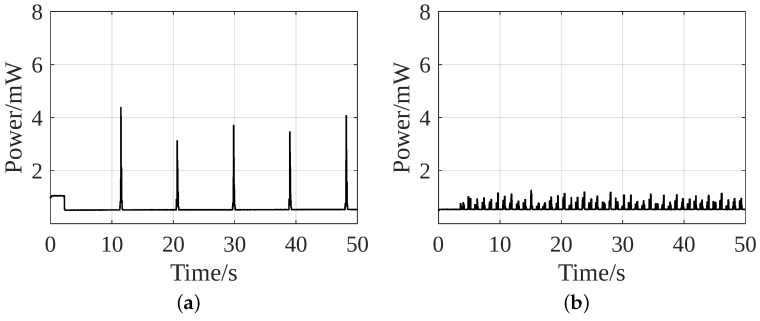
Power consumption of data acquisition and processing of (**a**) Strategy 1 and (**b**) Strategy 2.

**Figure 13 sensors-24-02549-f013:**
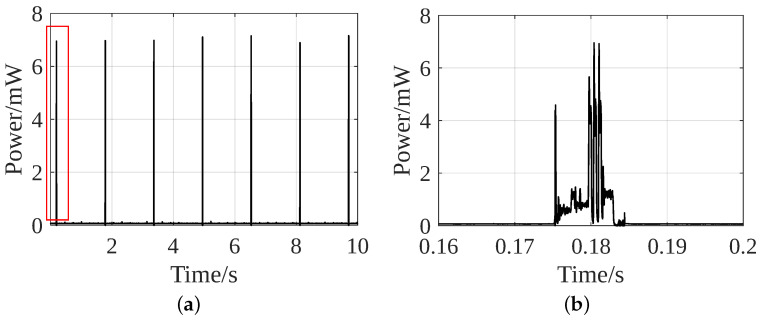
Power consumption of data transmission: (**a**) Strategy 1 and 2; (**b**) first advertising instant.

**Figure 14 sensors-24-02549-f014:**
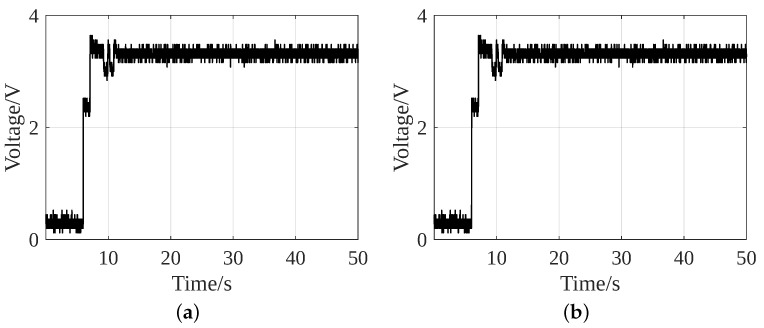
Output of LTC3331 with an additional battery of (**a**) Strategy 1 and (**b**) Strategy 2.

**Figure 15 sensors-24-02549-f015:**
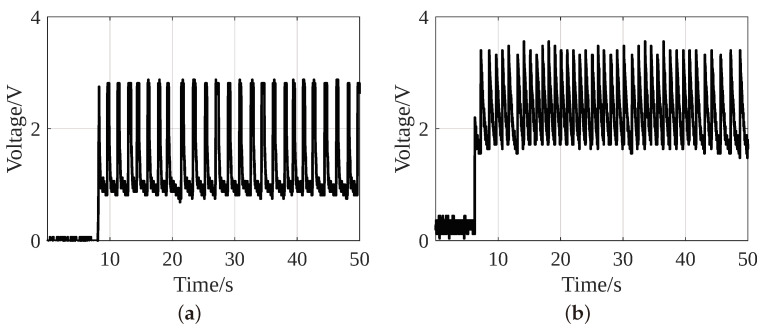
Output of LTC3331 without an additional battery of (**a**) Strategy 1 and (**b**) Strategy 2.

**Table 1 sensors-24-02549-t001:** Accuracy of activity recognition with the accelerometer.

Walk		Run		Accuracy
**Actual**	**Recorded**	**Actual**	**Recorded**	
20	20	20	19	97.5%
20	20	20	19	97.5%
50	49	50	47	96%
50	52	50	48	96%
50	50	50	51	99%
100	94	100	95	94.5%
100	97	100	99	98%
100	105	100	97	96%

**Table 2 sensors-24-02549-t002:** Accuracy of activity recognition with a piezoelectric sensor.

Walk		Run		Accuracy
**Actual**	**Recorded**	**Actual**	**Recorded**	
20	16	20	25	77.5%
20	15	20	23	80.0%
50	46	50	56	90.0%
50	46	50	58	88.0%
50	45	50	60	85.0%
100	84	100	113	85.5%
100	84	100	123	80.5%
100	90	100	121	83.5%

**Table 3 sensors-24-02549-t003:** Power consumption of Strategy 1 and 2.

Strategy		Strategy 1	Strategy 2
Average Accuracy (%)		96.8	83.75
Data acquisition and processing	Average power (mW)	0.56	0.66
Data acquisition and processing	Peak power (mW)	4.5	1.3
Data transmission	Average power (mW)	0.66	0.66
Data transmission	Peak power (mW)	7.3	7.3
Total	Average power (mW)	1.22	1.32
Total	Peak power (mW)	11.8	8.6
Steps to enable all functions		8–9	8–9
Peak charging current (μA/step)		67	75

## Data Availability

The original contributions presented in this study are included in the article; further inquiries can be directed to the corresponding authors.

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
