# Peer review of "Choice of Piezoelectric Element over Accelerometer for an Energy-Autonomous Shoe-Based System"

_sensors, 2024, doi:10.3390/s24082549_

Round 1

Reviewer 1 Report

Comments and Suggestions for Authors

This paper proposed a system for gait measurement with the help of PZT energy harvesters. The work is interesting, but some problems are not well solved. More works are needed to improve the system. The following is my main comments.

1. The paper intends to use PZTs as a source to power the system. However, the features of PZT harvesters are not tested and the output voltage and power is not given. Usually, the PZTs have an output power less than 1mW, and varies with the input stimulations. It is necessary to measure the PZT output features under the foot activities to show the feasibility of using PZTs to power the systems.

2.The information of PZT elements is missing, e.g. model, dimensions and the collection methods between each other.

3.It is needed to show a photo for the real system prototype.

4.In section 2, an illustration figure for the whole system construction is needed.

5.The paper title said the system is “Energy-Autonomous”, but a battery is needed according to the paper. A gait measurement cycle without the help from battery is needed. (Maybe, the battery should not appear in the whole if you want proposed a real self-powered system) .

6. I cannot well get the value of use the PZT as a sensing unit in Strategy 2. Limited by its ability, not all activities can be captured, and the task of gait monitoring is not well finished.

7. PZTs are fragile. Their viability under different loading conditions, e.g. weight of volunteers and intensity of activities, should be tested. Also. An expression about the efforts to improve the viability of PZTs should be added (Or a discussion). 

Author Response

Dear Reviewer,

Thank you for taking out your precious time to review this article. Your valuable comments have improved the article. Please find the response to your comments/suggestions below. The corresponding changes are highlighted in the resubmitted file.

This paper proposed a system for gait measurement with the help of PZT energy harvesters. The work is interesting, but some problems are not well solved. More works are needed to improve the system. The following is my main comments.

  1. The paper intends to use PZTs as a source to power the system. However, the features of PZT harvesters are not tested and the output voltage and power is not given. Usually, the PZTs have an output power less than 1mW, and varies with the input stimulations. It is necessary to measure the PZT output features under the foot activities to show the feasibility of using PZTs to power the systems.

The authors would like to acknowledge your valuable comment. The authors agree that including the gait signal in the article would offer an in-depth insight to the study. The power of a gait signal is added in subsection Strategy 2 (section 3.2.1). It was measured at a preliminary stage of research before the sole design and energy harvesting circuits were implemented.

  1. The information of PZT elements is missing, e.g. model, dimensions and the collection methods between each other.

Thank you for pointing out this missing information. The chosen piezoelectric element is shown in Figure 2. The collection method is already mentioned in section 2.4. The authors have used five elements for energy harvesting (denoted by 1,2,3,4 and 5) and one for sensing (denoted by S). The position of each element is chosen in such a way that as the foot strikes the ground, all of them also strike the ground. A two-layer sole layout is also designed with silicone rubber to embed the elements (Figure 3 and 7(a)). The sole layout provides a protection from wear and tear of the elements and their wires. It also increases the electromechanical conversion.

  1. It is needed to show a photo for the real system prototype.

Thank you for your suggestion. With response to this, the authors have added the real system prototype in Figure 7(b).

  1. In section 2, an illustration figure for the whole system construction is needed.

Thank you for your suggestion. The illustration of the whole system design is already shown in Figure 1. In alignment to the above comment, the authors have added Figure 6 and 7(a) to provide an overview about the system design. The authors hope that the latest addition has improved the illustration of the proposed system.

  1. The paper title said the system is “Energy-Autonomous”, but a battery is needed according to the paper. A gait measurement cycle without the help from battery is needed. (Maybe, the battery should not appear in the whole if you want proposed a real self-powered system).

Thank you for your comment.

The authors would like to highlight that a battery in Strategy 2 acts as a storage capacitor to ensure constant supply. The system is designed in a way that the extra harvested energy is charging the battery at the same time (line 359).  Thus, the battery does not require any external power to charge itself, which brings the concept of energy autonomous.

As Strategy 2 achieved the minimal operating voltage of MSP430, i.e. 1.8V, the fluctuating output without the battery in Figure 15(b) delivers sufficient power to obtain the desired feature. A relevant discussion is added in the last paragraph of section 4 (lines 325-337).

It is mentioned that one element delivers about 1.2mW peak power and therefore, with 5 such elements about 5-6 mW can be generated. This is sufficient for data acquisition and processing in both the strategies (In Table 3, it is mentioned as 4.5mW for Strategy 1 and 1.3mW for Strategy 2). However, the total peak power consumption of Strategy 1 and 2 are 11.8mW and 8.6mW, where Bluetooth data transmission power consumption is very high (7.3mW in both strategies as same module is used). In each strategy, the difference of peak power and power generated by the 5 piezoelectric elements is derived from the battery. As mentioned above, this battery is charged during walking and not from external power supply. The peak power demand is only for 0.02s during advertising event and the battery is used for only that very small period.

Henceforth, there are two possible options – 1) use UART module to make it “battery-less” energy autonomous or 2) rely on battery for the additional demand during Bluetooth data transmission.

For Bluetooth transmission, assumption that piezoelectric energy harvesting is delivering 6mW, Strategy 1 needs an extra 5.8mW while Strategy 2 needs 2.6mW only. In such scenario, the battery acts as a backup for urgent need for the 0.02s advertising event.

  1. I cannot well get the value of use the PZT as a sensing unit in Strategy 2. Limited by its ability, not all activities can be captured, and the task of gait monitoring is not well finished.

The authors appreciate your feedback and would like to express their approach with more clarity.  

A piezoelectric element has exciting qualities suitable for its sensing application, e.g. plantar pressure map as discussed in the subsection 1.1. As the title implies, the article explores the potential of piezoelectric element over accelerometer in an energy autonomous shoe system. In the initial submission, the accelerometer recognizes 3 activities while piezoelectric element recognizes 2 activities. The jump activity can also be derived from piezoelectric element using ADC of MSP430 but is intentionally avoided to lower down the power consumption of the system.

From this comment, the authors have figured out that addition of jump activity could possibly create confusion to readers (if accepted for publication). Therefore, to keep fair comparison in both strategies, the jump activity of strategy 1 has been dropped.  Now, both strategies only consider run and walk. We express our gratitude to the reviewer once again.

The authors present a power optimized method with piezoelectric element as sensor and energy harvester in one system. The accelerometer needs external power to operate while the piezoelectric element operates on the foot pressure. As the aim is to make energy-autonomous, only step-counting and two activity recognition (walk and run), is taken into consideration. It is to be noted that the piezoelectric sensors can capture same activities as an accelerometer. However, the Strategy 2 is proposed in such a way that the energy harvested from walking can deliver enough output to operate its mentioned functionalities. It presents a comprehensive study to take advantage of interesting materials like piezoelectric elements, which can act as sensor and energy harvester at the same time. For trending lifestyle wearables where accuracy can be compromised, with the interests in sustainable energy harvesting and risks of battery disposals, piezoelectric elements is an interesting choice.

  1. PZTs are fragile. Their viability under different loading conditions, e.g. weight of volunteers and intensity of activities, should be tested. Also. An expression about the efforts to improve the viability of PZTs should be added (Or a discussion). 

The authors completely agree that PZTs are fragile. The material needs to be investigated more, especially from the perspective of ambient energy harvesting. The energy harvested from such elements increases with increase in mechanical vibration. The challenge is that human walking is only a few hertz, 1-3 Hz depending on the type of activity.

At the preliminary stage of this research, weight of volunteers and different activities have been tested. However, with the proposed sole design and specific elements used, these factors did not make significant differences. The energy harvested depends on the electromechanical conversion, which is determined by type of material and sole layout (or any other hosting structure for different application) where the elements are embedded. As material synthesis is beyond expertise and experience, the authors have focussed on low-cost, light-weight commercial elements. Within the scope of authors, a silicon rubber sole layout is designed to ensure comfort to the subject and low power electronic components are chosen. The microcontroller (MSP430FR5947) and voltage regulator (LTC3331) implemented in the study have minimal power consumption. Sole design is crucial for its application. Thus, the article proposed a power optimized method for an energy autonomous shoe system. In support of this comment, a paragraph is added in the conclusion (lines 366-374).

Reviewer 2 Report

Comments and Suggestions for Authors

In this paper, both sensing and energy harvesting together embedded in a shoe sole are investigated for two different strategies i.e., an accelerometer as a sensor for gait acquisition and a piezoelectric element as an energy harveser for the same. The study is interesting due to considering the simultaneous sensing and energy harvesting. The article is well organized. My suggestion is that the authors should give the power equation and illustrate how the power outputs are calculated for a fair comparision since the power is an important evaluation index for two different strategies.

Author Response

In this paper, both sensing and energy harvesting together embedded in a shoe sole are investigated for two different strategies i.e., an accelerometer as a sensor for gait acquisition and a piezoelectric element as an energy harveser for the same. The study is interesting due to considering the simultaneous sensing and energy harvesting. The article is well organized. My suggestion is that the authors should give the power equation and illustrate how the power outputs are calculated for a fair comparision since the power is an important evaluation index for two different strategies.

Dear Reviewer,

Thank you for taking out your precious time to review this article. Your valuable comment has improved the article. Please find the response to your comments/suggestions below. The corresponding changes are highlighted in the resubmitted file.

A new subsection Power Calculation is added (section 3.3) in agreement to your valuable comment. The power is the product of current measure by NRF power profiler kit and operating voltage 3.3V.

Reviewer 3 Report

Comments and Suggestions for Authors

The authors of the paper "Choice of Piezoelectric Element over Accelerometer for an Energy-Autonomous Shoe-based System" have produced an article that meets all the written and unwritten rules and is in perfect agreement with the MDPI template. The approach is interesting from the point of view of energy harvesting and IoT application delivery via Bluetooth from the perspective of experimental results. I read this article carefully and as it rarely happens; I have nothing to recommend for improvement, the article is very carefully written. Absolutely by way of optional recommendation it should perhaps be better highlighted in the introduction what is original in this approach other than the destination. There are no new elements to the concept or electronic devices used. On the other hand it is a good example carefully elaborated and described in sufficient detail for replication, thus being an example with tutorial value. This last element leads me to consider this article ready for publication.

Author Response

The authors of the paper "Choice of Piezoelectric Element over Accelerometer for an Energy-Autonomous Shoe-based System" have produced an article that meets all the written and unwritten rules and is in perfect agreement with the MDPI template. The approach is interesting from the point of view of energy harvesting and IoT application delivery via Bluetooth from the perspective of experimental results. I read this article carefully and as it rarely happens; I have nothing to recommend for improvement, the article is very carefully written. Absolutely by way of optional recommendation it should perhaps be better highlighted in the introduction what is original in this approach other than the destination. There are no new elements to the concept or electronic devices used. On the other hand it is a good example carefully elaborated and described in sufficient detail for replication, thus being an example with tutorial value. This last element leads me to consider this article ready for publication.

Dear Reviewer,

Thank you for taking out your precious time to review this article. Your valuable comments have improved the article. Please find the response to your comments/suggestions below. The corresponding changes are highlighted in the resubmitted file.

The abstract and proposed work has been restructured to bring clarity to the concept. Please find the modifications in abstract and section 1.3.

Round 2

Reviewer 1 Report

Comments and Suggestions for Authors

Thank you for the great responses. However, I’m still interested in the energy from PZTs.

1. In Fig.10, a result for the output power of PZT is given. But how to get this results, e.g. the equipment, methods and measuring condition?

2. What’s the type of utilized PZT? PZT-5 or someone else?

3. From Fig.1, the battery and MCU all acts as a load for LTC chip. How to connect the battery to the output of LTC chip? How to decide the direction of energy flow? Why the output voltage of LTC chip in Fig.14 is much stable and higher than the one in Fig. 15 when the system is used to power an additional load?  Then, a charging process of an empty battery during the activity measurement should be shown to show the ability of PZTs as a power source.

4. A video for the whole working process in the measurements can be given to realize a better way to show the results (especially the differences with & without a battery).

5. The information and photo for the battery is not given in the paper. They should be added.

6. What are the numbers in Table 1 and 2? The repeating times for each activity?

7. Is it possible to recognize the type of activity according to the output of sensing elements in both Strategies?

8. How long does it take from the start of activity to receiving the desired data when with & without the battery?

Author Response

Dear Reviewer,

Thank you for taking out your to review this article. Please find the response to your comments/suggestions below. The corresponding changes are highlighted in the resubmitted file.

  1. In Fig.10, a result for the output power of PZT is given. But how to get this results, e.g. the equipment, methods and measuring condition?

The authors would like to acknowledge your valuable comment.

Figure A: Gait voltage signal

The gait signal shown in Figure 10 is based on voltage signal recorded in oscilloscope. Following steps are used to estimate the power -

  1. Initially, the voltage signal is recorded in oscilloscope (shown above Figure A).
  2. For each peak voltage, the current delivered is about 20-25 µA. Based on this measurement, resistance is calculated i.e.

This leads to an approximate Resistance value of 2-2.5MΩ. By its material property, a PZT has a very high intrinsic resistance as result of which it delivers low current.

  1. Figure 10 is only a power calculation based on above assumptions. The power is calculated based on , where voltage is recorded in oscilloscope and resistance is the intrinsic resistance of the PZT element, discussed above.

  1. What’s the type of utilized PZT? PZT-5 or someone else?

The authors would like to thank your valuable feedback.

The choice of PZT is already mentioned in the reference (76). The link is https://www.murata.com/en-eu/products/productdetail?partno=7BB-20-6.

On contacting the customer support of Murata at the initial phase of research, they informed us about PZT-5 only.

  1. The authors would like to acknowledge your valuable comment. The review comment “3” has been subcategorized as follows-

  1. From Fig.1, the battery and MCU all acts as a load for LTC chip. How to connect the battery to the output of LTC chip? How to decide the direction of energy flow?

The connection of the battery is shown below. You can find more information in the datasheet - https://www.analog.com/media/en/technical-documentation/data-sheets/3331fc.pdf

Figure B: LTC3331

b. Why the output voltage of LTC chip in Fig.14 is much stable and higher than the one in Fig. 15 when the system is used to power an additional load?  Then, a charging process of an empty battery during the activity measurement should be shown to show the ability of PZTs as a power source.

It is important to state that the output of LTC3331 is set to 3.3V. The output voltage of LTC3331 is stable (but not higher) in Fig 14 than Fig 15. Only the output voltage in Fig 15(a) is lower than Fig 14(a), 14(b) and 15(b) i.e. 3.3V.  

The reason is because battery acts as a backup storage which results in stability. This has been already discussed in lines 322-325.

Figure 15(b) i.e. Strategy 2 without additional battery can however achieve the desired voltage. Absence of battery is the reason behind instability. Thus, the main conclusion from this study is the battery acts as a backup and buffer capacitor.

The authors would like to emphasise that the focus of this article is Strategy 2. It is a power optimized technique that takes advantage of simultaneous sensing and energy harvesting.  

We have mentioned in Table 3 already the measured charging current i.e. 67uA/step for Strategy 1 and 75uA/step in Strategy 2. This also brings to conclusion that Strategy 2 is better than Strategy 1 in charging the battery.

  1. A video for the whole working process in the measurements can be given to realize a better way to show the results (especially the differences with & without a battery).

The authors respects your suggestion.

It would have been a great addition to this article. However, we regret to inform you that the authors are no longer in a state to record a video as mentioned above. Those who have conducted the experiments have already graduated and moved to different places of the world.

However, there are other old videos of working system, which has been attached in the report.

  1. The information and photo for the battery is not given in the paper. They should be added.

The authors would like to thank you for pointing this out. It has been missed out in the previous submission. Figure 6 has been edited accordingly. Please check block 8.

  1. What are the numbers in Table 1 and 2? The repeating times for each activity?

The authors would like to acknowledge your comment.

Yes, you are right. Each activity is repeated and then accuracy is measured.

E.g. 20 steps are taken. The person documents manually i.e. actual and observes recorded data in the mobile phone i.e. recorded.

  1. Is it possible to recognize the type of activity according to the output of sensing elements in both Strategies?

The authors appreciates your valuable comment.

In the article, all activity recognition is based on sensor output in both strategies. The subsections 3.1 and 3.2 have mentioned about the sensor data information and the approaches taken to recognize the type of activity.

  1. How long does it take from the start of activity to receiving the desired data when with & without the battery?

The authors would like to appreciate your valuable comment.

The article considered number of steps as parameter to achieve the desired objective. With battery, both strategies take 8-9 steps for a healthy adult. As each step takes about 1 second, so 8-9 steps imply 9 seconds overall.

Without battery, Strategy 1 fails to achieve the desired objective (ref Fig 15a) but Strategy 2 takes 8-9 steps i.e. 9 seconds.
